# Impact of the COVID-19 pandemic on the routine immunization system in Bangladesh

**Md. Tanvir Hossen**[1,2], **Md Sayik Bin Alam**[2], **Md. Sahidul Islam**[3], **Emanuele Montomoli**[4], **Ralf Clemens**[1,5,6,7], **Sue Ann Costa Clemens**[1], **Mohammad Delwer Hossain Hawlader**[8], **Md Foyjul Islam**[9]*

1 Institute of Global Health IFGH, University of Siena, Siena, Italy, 2 Directorate General of Health Service, Dhaka, Bangladesh, 3 World Health Organization, Dhaka, Bangladesh, 4 Department of Molecular and Developmental Medicine, University of Siena, Siena, Italy, 5 BMGF, Seattle, United States of America, 6 International Vaccine Institute (IVI), Seoul, South Korea, 7 Curevac AG, Tuebingen, Germany, 8 Department of Public Health, North South University, Dhaka, Bangladesh, 9 Department of Epidemiology, Institute of Epidemiology, Disease Control and Research (IEDCR), Dhaka, Bangladesh

* drislam0666@gmail.com

## Abstract

The COVID-19 pandemic has put immense pressure on countries' health structures to maintain routine immunization and VPD surveillance programs, especially in LMICs such as Bangladesh. Understanding the effects of COVID-19 will allow countries like Bangladesh to become better prepared for future health emergencies, so this study explored the effects of COVID-19 on routine immunization and VPD surveillance programs in Bangladesh. With a sequential mixed-method approach, the quantitative data was collected from the DHIS2 for January 2019-December 2021. The qualitative data was collected from ten KIIs, which was conducted with the key stakeholders of EPI and VPD surveillance in Bangladesh. Findings suggest that there had been three drop-rebound periods in the routine immunization program in Bangladesh. The study identified that Bangladesh was able to catch up with the pre-pandemic level immunization within four to six months from the first drop, and the immunization drops became less severe with the progression of time. COVID-19-related movement restrictions, lack of workforce, fear and concern regarding COVID-19, prioritizing COVID-19 vaccination, lack of a comprehensive EPI structure in urban compared to rural areas, and lack of knowledge to conduct EPI and VPD activities amidst the pandemic situation were identified as the main reasons for these drops.

## Introduction

Childhood vaccination is one of the most important and cost-effective public health interventions to ensure protection against health maladies and reduce child mortality [1–4]. In recent years, significant attention has been paid to promoting and expanding vaccination initiatives to increase coverage and reduce the burden of

**Data availability statement:** The data used in this study were obtained from the Management Information System (MIS) Unit of the Directorate General of Health Services (DGHS), Bangladesh. These data are not publicly shareable due to legal and ethical restrictions related to patient confidentiality and institutional data-sharing policies. Specifically, the dataset contains sensitive health service utilization information linked to administrative units and is governed by national surveillance confidentiality regulations. Access to the data is restricted by the DGHS. Interested researchers may request access to the dataset by contacting the DGHS MIS Unit via email at: mis@dghs.gov.bd.

**Funding:** The author(s) received no specific funding for this work.

**Competing interests:** The authors have declared that no competing interests exist.

vaccine-preventable disease-based mortality and morbidity, especially in accordance with the Immunization Agenda 2030.

This immunization agenda can be thought of as a continuation of the original Expanded Program on Immunization (EPI) by the World Health Organization initiated back in 1974 and a roadmap for the future of immunization in the upcoming decade [1,2]. The EPI program focused on ensuring access to four vaccines (BCG, DTP, Pol, and MCV) for all children around the world to protect them from six primary vaccine-preventable childhood diseases, i.e., tuberculosis, diphtheria, tetanus, pertussis, poliomyelitis, and measles [5]. With the progression of time, more vaccines and doses have become included in the EPI program to broaden the protection further [1,2]. Studies show that these EPI interventions have played the most crucial role in reducing child mortality globally, especially in low-middle-income countries (LMICs), and the Immunization Agenda 2030 would further strengthen immunization initiatives around the world to reduce mortality and morbidity [6].

However, the COVID-19 pandemic, caused by the coronavirus (SARS-CoV-2), has caused significant disruption in routine immunization programs all over the world, especially in LMICs [7–13]. WHO reported that mass immunization campaigns were temporarily disrupted in many countries due to the COVID-19 pandemic [14]. A modelling study showed an upward trend in mortality among children aged <5 years may have been encountered resulting from disrupted healthcare service in low- and middle-income countries [14]. The WHO, UNICEF, Gavi and the Sabin Vaccine Institute stated that the current pandemic may have caused a significant hindrance in the provision of routine immunization services in about 68 countries affecting approximately 80 million children under the age of 1 year living in these countries has yet to be detailed at the global level [14]. This evidence shows the fragility of immunization programs around the world, especially in the LMICs.

However, after the COVID-19 pandemic hit Bangladesh in March 2020, the government quickly started implementing countrywide COVID-19 control measures, e.g., social distancing, lockdown of areas, closure of educational institutions, work from home and limiting physical presence in government and non-government organizations and offices, from March 2020, which might have affected people's access to the existing routine immunization services in Bangladesh [15,16]. Studies conducted on this phenomenon either looked at the general disruption of the country's primary healthcare service delivery system or in a specific geographical area to find disruption in the immunization system [9,17–21]. In addition, these studies predominantly measured disruption based on vaccination coverage and vaccination session conduction-related data, whereas it is crucial to look at the vaccine-preventable disease (VPD) surveillance performance indicators as well. Moreover, it is equally important to know about the mitigation measures taken by the country to address the emerging challenges during the pandemic and their impact as well.

Since the country-specific detailed information is scarce, this study aims at the immunization system of Bangladesh with a view to capture and compare its vaccination coverage and vaccine-preventable disease surveillance performance indicators before and during the pandemic as well as to understand the measures taken by the

country to restore the potential disruptions. It also aims to provide recommendations on the implications of immunization and VPD surveillance system-related policy based on the study findings. With these objectives, this study attempts to provide the impact of the pandemic on vaccination coverage and equity (antigen-wise national coverage, gender-wise coverage, urban and rural vaccination coverage status, hard-to-reach area coverage, dropout rate, left-out rate, session planned and held) & VPD surveillance indicators (Measles-Rubella, Acute Flaccid Paralysis, Acute Encephalitis Syndrome & Neonatal tetanus surveillance) by comparing the scenario before and during the pandemic using the district information system 2 (DHIS2) database of the country alongside identifying the challenges and measure taken to restore the system.

## Materials and methods

This mixed-method study followed a nested design, in which the qualitative component was nested in the quantitative component. The data collection for this study was done in sequence, i.e., quantitative data was collected first, and then qualitative data was collected.

### Quantitative component

To assess the impact of the COVID-19 pandemic on routine immunization and vaccine-preventable disease (VPD) surveillance in Bangladesh, quantitative data were collected from national health information systems covering the period from January 2019 to December 2021. Immunization coverage data—focused on children aged 0–2 years and women aged 15–49 years—were obtained from the District Health Information Software 2 (DHIS2) on May 15, 2022. Surveillance data on VPDs were retrieved from the Excel-based reporting system maintained by the Expanded Program on Immunization (EPI). The primary exposure variable was defined as the occurrence of three major phases of the COVID-19 pandemic: March–April 2020 (first lockdown), December 2020 (second wave), and July 2021 (Delta variant surge). The main outcome variable was vaccination coverage, measured as the proportion of eligible children receiving key antigens, including Bacillus Calmette–Guérin (BCG), Pentavalent-1, Pentavalent-3, and Measles-Rubella first dose (MR1). Covariates included sex, geographic region, urban versus rural status, hard-to-reach area classification, dropout and left-out rates, and the number of planned versus conducted immunization sessions. VPD surveillance performance was assessed based on trends in reported cases of measles-rubella, acute flaccid paralysis, acute encephalitis syndrome, and neonatal tetanus. Exploratory data analysis techniques were applied to examine temporal trends and performance across the defined time periods. Descriptive statistics were used to summarize key indicators, with significance interpreted at a 5% level and 95% confidence intervals reported where applicable.

### Qualitative component

The qualitative data for this study was collected using Key Informant Interviews (KIIs) with various stakeholders based on their direct involvement in the Expanded Program on Immunization (EPI) and VPD surveillance, including national and subnational-level government stakeholders in urban and rural areas, core immunization non-government developmental partners such as UN organizations, particularly the World Health Organization and UNICEF, and front-line workers. The participants for the KIIs were selected using purposive sampling, and the sample size was determined to be 10 after data saturation was achieved. This approach aligns with qualitative research standards, where 6–12 interviews are often sufficient for studies with a relatively homogeneous group of experts [22].

The KIIs were conducted using an online platform, i.e., Zoom, based on a semi-structured KII guideline that was developed for this study-based literature review and brainstorming. The KII guideline was fine-tuned and modified based on the data gathered from the quantitative analysis. The participants were free to choose the language of the interview, and four out of the ten interviews were conducted in Bengali, while the rest were conducted in English.

All the recorded KIIs were transcribed and translated into English following a verbatim transcription technique, as it offers rich descriptions to understand the context. All the transcriptions then went through a de-identification process, in which, any and every information related to the participant's identity was modified and pseudonym was used where necessary. Interviews were also assigned with an arbitrary naming system as part of the de-identification process.

All the de-identified transcripts were analyzed using a six-stage thematic analysis strategy with the Atlas. Ti (version 22.1.0) software package [23]. All the audio recordings were de-identified using pseudonyms and stored in a secure location that can only be accessed by the researcher.

### Ethical considerations

This study was conducted in accordance with the ethical principles outlined in the Declaration of Helsinki. Ethical approval was obtained from the Institutional Review Board (IRB) of North South University, Dhaka, Bangladesh (Memo No. 2022/OR-NSU/IRB/0406). Written informed consent was obtained from all participants involved in the qualitative component (key informant interviews). Participation was voluntary, and the confidentiality of the respondents was strictly maintained. No personal identifiers were collected during data analysis or reporting. The quantitative data used in this study were extracted from the national DHIS2 database with appropriate permission from the relevant authorities.

## Results

### Vaccination coverage before and during the pandemic

The findings of this study suggest that vaccination coverage went through three drop-rebound phases during the COVID-19 pandemic in Bangladesh before completely returning to the pre-pandemic level in December 2021. The first drop-rebound phase was identified during the first wave of COVID-19 in Bangladesh (March-April 2020), when vaccination coverage for almost all the vaccines dropped close to 55% (Fig 1).

Despite the ongoing pandemic and lockdown restrictions, the vaccination coverage slowly rebounded for the next six months (June-Nov 2020) before dropping again in December 2020, when the second wave of the COVID-19 pandemic arrived and associated lockdown measures were put into place. For the next six months (Jan-Jun 2021), the coverage fluctuated but maintained the rebound trend. It dropped again in July 2021, which is the third and final drop-rebound phase for the COVID-19 pandemic in Bangladesh (Fig 1).

While explaining the reason behind such a sharp decline in vaccination coverage during that time, one of the respondents explained,

*"At the beginning of the pandemic, we had 70 consecutive days of lockdown… when all communications were cut off on March 24, 2020, [it] was a tough time for us. Almost all the employees went to their respective homes… since people did not go out of their houses, our city corporation vaccination centers were almost closed, but our upazila-level vaccination donation centers were open. People did not come there because if a COVID-19 patient were identified, then the entire upazila would have been locked down."*

Quotation 1 [KII8, Government High Official]

These seventy consecutive days of lockdown began at the end of March. They lasted almost the second half of June, during which there was a strict nationwide restriction of movement for people and a ban on mass gatherings to curb the spread of infection during the first wave. These restrictions on movement and mass gatherings during the COVID-19 pandemic meant that people could not move to vaccination centers or be allowed to gather there. In addition, if individuals tested positive for COVID-19, the government locked down entire communities and upazilas, which means vaccination activities could not be continued in that area for at least two weeks. On top of that, many vaccinators were affected; one of the respondents mentioned,

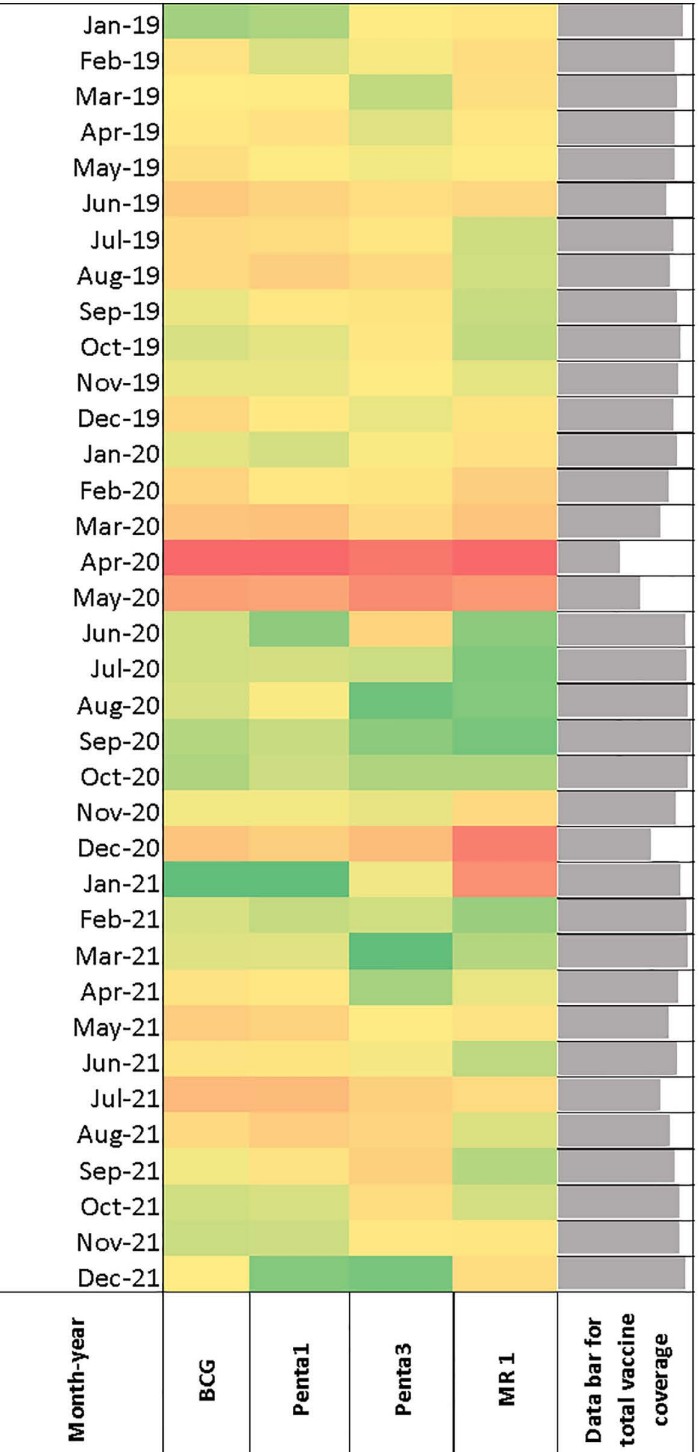
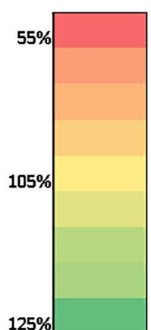

**Fig 1. Vaccination Coverage before and during the COVID-19 pandemic in Bangladesh.**

*"What I would say at the field level is that in many cases, our staff had been affected, so they had to maintain isolation and quarantine when they had tested positive. So, it turned out that on the date we were supposed to vaccinate at a center, maybe we had to drop that center because we did not have the workforce, and because of the lack of workforce, we will not be able to do the vaccination, in that case, we had to wait for the next date for that center. My staff will come back, and we will work again. I had noticed some of these difficulties."*

Quotation 2 [KII7, UH&FPO]

However, one of the respondents argued,

*"The first drop at the beginning of COVID was serious, but the second drop, I am not sure if we can call that a drop, as the issues with MR vaccination during that time were caused by data gathering issues. There was a campaign going on for MR vaccination, and we have vaccinated almost all children under seven months. However, the data of this vaccination was calculated with the campaign, not with the national data… the third drop was caused by the emergence of the delta variant and the beginning of COVID vaccination. The government utilized almost all workforce and resources to start and complete COVID vaccination effectively, which affected the EPI vaccination efforts a bit."*

[KII2, UN Official]

## BCG vaccine coverage before and during the COVID-19 pandemic

Findings suggest that the average BCG vaccination coverage began to improve in June 2020 and consistently exceeded the pre-pandemic level coverage in October 2021 (Table 1). An in-depth exploration of the data found that, although the vaccination coverage fluctuated during the pandemic, it never went down to the early COVID-19 pandemic level (March-April-May 2020) in Bangladesh; rather, the vaccination coverage crossed the pre-pandemic level consistently for three months twice, first in January-March 2021 and then in October-December 2021.

**Table 1. Average BCG vaccine coverage (%) during 2019 to 2021 in Bangladesh.**

| Months | Average BCG vaccine coverage (%) | | | | | |
|---|---|---|---|---|---|---|
| | % Change in coverage (95% CI) | | | | | |
| | 2019 | 2020 | 2021 | 2019–2020 | 2019–2021 | 2020–2021 |
| January | 116 | 106 | 125 | −9 (−10.5, −7.5) | 9 (8.5, 9.5) | 18 (15.9, 20.1) |
| February | 99 | 94 | 108 | −5 (−6.5, −3.5) | 9 (8.5, 9.5) | 15 (12.9, 17.1) |
| March | 102 | 88 | 107 | −14 (−15.5, −12.5) | 5 (4.5, 5.5) | 22 (19.9, 24.1) |
| April | 101 | 55 | 99 | −46 (−47.5, −44.5) | −2 (−2.5, 1.5) | 80 (77.9, 82.1) |
| May | 98 | 75 | 91 | −24 (−25.5, −22.5) | −7 (−7.5, −6.5) | 21 (18.9, 28.3) |
| June | 90 | 109 | 99 | 21 (19.5, 22.5) | 9 (8.5, 9.5) | −9 (−11.1, −6.9) |
| July | 96 | 109 | 84 | 14 (12.5, 15.5) | −12 (−12.5, −11.5) | −23 (−25.1, −20.9) |
| August | 96 | 108 | 96 | 13 (11.5, 14.5) | 0 (−.5,.5) | −11 (−13.1, −8.9) |
| September | 105 | 113 | 104 | 8 (6.5, 9.5) | −1 (−1.5,.5) | −8 (−10.1, −5.9) |
| October | 108 | 114 | 109 | 6 (4.5, 7.5) | 1 (−.5, 1.5) | −5 (−7.1, −2.9) |
| November | 105 | 104 | 110 | −1 (−2.5, 0.5) | 5 (4.5, 5.5) | 6 (3.9, 8.1) |
| December | 95 | 88 | 102 | −7 (−8.5, −5.5) | 7 (6.5, 7.5) | 16 (13.9, 18.1) |

**Footnote:** Coverage levels above 100% reflect limitations in the denominator used for coverage calculation, as aggregated reporting often relies on approximate target populations. This can lead to overestimation, consistent with previous findings that adjusting denominators reveals discrepancies with standard reporting methods [23].

Table 2. Change in average coverage of Penta 1, Penta 3, and MR1 vaccines between 2019-2021.

| Months | Penta 1 vaccine coverage | | Penta 3 vaccine coverage | | MR1 vaccine Coverage | |
|---|---|---|---|---|---|---|
| | % Change in coverage (95% CI) | | | | | |
| | 2019 - 2020 | 2019 - 2021 | 2019 - 2020 | 2019–2021 | 2019 - 2020 | 2019 - 2021 |
| January | −6 (−7.1, 4.9) | 11 (10.3, 11.7) | 1 (−.8, 2.8) | 2 (1.4, 2.6) | −5 (−6.6, −3.4) | −37 (−38.2, −35.8) |
| February | −7 (−8.1, −5.9) | 4 (3.3, 4.7) | −4 (−5.8, −2.2) | 6 (5.4, 6.6) | −4 (−5.6, −2.4) | 18 (16.8, 19.2) |
| March | −15 (−16.1, −13.9) | 6 (5.3, 6.7) | −14 (−15.8, −12.2) | 13 (12.4, 13.6) | −13 (−14.6, −11) | 18 (16.8, 19.2) |
| April | −44 (−45.1, −42.9) | 3 (2.3, 3.7) | −48 (−49.8, −46.2) | 8 (7.4, 8.6) | −45 (−46.6, −43.4) | 2 (.8, 3.2) |
| May | −25 (−26.1, −23.9) | −9 (−9.7, −8.3) | −39 (−40.8, −37.2) | −1 (−1.6, −.4) | −23 (−24.6, −21.4) | 3 (1.8, 4.2) |
| June | 29 (27.9, 30.1) | 7 (6.3, 7.7) | −3 (−4.8, −1.2) | 7 (6.4, 7.6) | 26 (24.4, −27.6) | 21 (19.8, 22.2) |
| July | 14 (12.9, 15.1) | −12 (−12.7, −11.3) | 9 (7.2, 10.8) | −9 (−9.6, −8.4) | 11 (9.4, 12.6) | −7 (−8.2, −5.8) |
| August | 12 (10.9, 13.1) | −1 (−1.7, −.3) | 30 (28.2, 31.8) | −2 (−2.6, −1.4) | 7 (5.4, 8.6) | 3 (1.8, 4.2) |
| September | 10 (8.9, 11.1) | −2 (−2.7, −1.3) | 20 (18.2, 21.8) | −8 (−8.6, −7.4) | 6 (4.4, 7.6) | 2 (.8, 3.2) |
| October | 4 (2.9, 5.1) | 2 (1.3, 2.7) | 14 (12.2, 15.8) | −4 (−4.6, −3.4) | 2 (.4, 3.6) | 0 (−1.2, 1.2) |
| November | −2 (−3.1, −.9) | 5 (4.3, 5.7) | 4 (2.2, 5.8) | −1 (−1.6, −.4) | −5 (−6.6, −3.4) | 1 (−.2, 2.2) |
| December | −10 (−11.1, −8.9) | 21 (20.3, 21.7) | −21 (−22.8, −19.2) | 16 (15.4, 16.6) | −40 (−41.6, −38.4) | −3 (−4.2, −1.8) |

District-level data further strengthens these findings, as it shows that the average BCG vaccination coverage improved in most of the districts in 2021 compared to the pre-pandemic level (Table 2) The highest improvement of coverage occurred in the three periphery districts, i.e., Naogan, Rangpur, and Cox's Bazar, while the highest decrease in coverage occurred in Dhaka and Meherpur.

While explaining the reason behind improved BCG vaccination coverage all over the country but decreased coverage in the capital, one of the KII respondents previously mentioned the issue of prolonged closure of vaccination centers in the city corporation areas while the vaccination centers in the rural areas remained open (Quotation 1), another respondent, who is working in the Dhaka City Corporation, said:

*"[i]n rural areas, the vaccinators are mostly from those localities, and they know every house. However, the scenario is very different in urban areas. Some floating people may have taken the BCG-1 or Penta-1 From Dhaka and then migrated to the rural areas or other districts due to the COVID-19 lockdown or other reasons. That is why there are some lickings [in coverage] … the mothers of the urban area are mostly working women, so they often cannot bring their child in the daytime for vaccination."*

Quotation 3 [KII1, Health Care Manager, Dhaka]

Moreover, the respondent went on to say:

*"You know, in Bangladesh, the Rural Health care facility is much more organized than that of this Urban area because, in the urban area, there are no such ladders [hierarchy of health facilities] like Union sub-center at the base, then Upazila Health Complex and then others. [In Dhaka] we have only five CRNCCs, which are run by some NGOs; we do not have any of our human resources to run all those, so we have to depend on the NGOs, so some are lacking out there, but we are trying to overcome those."*

Quotation 4 [KII1, Health Care Manager, Dhaka]

This respondent mentioned multiple reasons to explain the decreased vaccine coverage in Dhaka city; these are: (i) the health structure is structured in the rural areas of Bangladesh compared to the urban areas, (ii) ease of tracking people for

the vaccinators in rural areas than in the urban areas, as the vaccinators in the rural area are locals but the vaccinators in urban area are not, (iii) the lack of workforce in an urban area and the dependency on NGOs for the workforce, (iv) migration of people to other areas after starting vaccination, especially, the floating people is hard to keep track as they do not have any permanent residents in the urban areas, (iv) the clash of vaccination centers opening hours with working mothers work timetable because of which they often cannot bring their child at the vaccination centers. Other KII respondents also frequently mention these issues, especially emphasizing the issue of floating people and working mothers.

### Penta 1, Penta 3, and MR1 Vaccine Coverage before and during the COVID-19 pandemic

The Penta 1 and Penta 3 vaccination coverage before and during the COVID-19 pandemic is very similar to the average BCG vaccination coverage of that timeframe (Table 2). Change in average coverage of Penta 1, Penta 3, and MR1 vaccines between 2019–2021), as the average vaccination coverage for both these vaccines surpassed the pre-pandemic level twice in 2021, i.e., at the beginning and the end of 2021.

However, vaccination coverage for MR1 vaccines showed an opposite trend, as it decreased at the beginning and at the end of 2021 while improving almost in all other months.

The study found that the vaccination coverage for Penta 1, Penta 3, and MR1 changed positively between 2019 and 2021. However, the Penta 1 and Penta 3 vaccination coverage increased the most in Rangpur, followed by Cox's Bazar. On the other hand, the MR1 coverage increased the most in Naogaon, followed by Narail.

On the other hand, the highest decrease of vaccination coverage between 2019 and 2021 was observed in Dhaka for Penta 1, in Dhaka and Brahmanbaria for Penta 3, and in Madaripur for MR1. The MR1 vaccination coverage experienced a less turbulent change among these three vaccines, as its decrease in coverage mostly remained around three percent, while the decrease in coverage fluctuated most for Penta 1 (Fig 2).

This study analyzed the trend of BCG, PENTA 1, PENTA 3 and MR 1 coverage from 2019 to 2021. Coverage for all the vaccines in 2019 and 2021 was found to be almost similar throughout the months, whereas coverages for 2020 were found to be far below from March to April compared to 2019 and 2021. From May 2020 onwards, coverages started to catch up (Fig 2).

### Urban-Rural dynamics of EPI coverage

This study also explored the dynamics of urban-rural EPI coverage and found that it was slightly better in rural areas than in urban areas.

This study compared coverages of different antigens between the urban and rural areas across the three years of 2019–2021. The coverage of BCG, PENTA 1, PENTA 3, and MR 1 varied exceptionally. However, in 2019, due to the development, approval, and initiation of implementation of the National Urban Immunization Strategy, the coverage in the urban areas may have improved, which dropped significantly in the following year 20,20. In addition, in the following year, 2021, coverage in the urban areas for all four antigens was slow to catch up compared to the rural areas (Fig 3).

### VPD surveillance performance indicators before and during the pandemic

Evaluating the performance of the Vaccine Preventable Disease (VPD) surveillance is one of the objectives of this study. While comparing the VPD surveillance performance indicators before and during the pandemic, it was found that the surveillance program was significantly affected by the COVID-19 pandemic.

The surveillance for AES, AFP, and MR plunged in April 2020 and then began to improve gradually in the later months. For the AES, the surveillance went back to the pre-pandemic level in December 2020, and for AFP, it was around July-August 2020. The surveillance for MR reflected the trend of 2019 in both 2020 and 2021, as the surveillance went down in the second and fourth quarters. Interestingly, the surveillance for CRS remained mostly steady and upward in 2020, performing even better than in 2019 and 2021 (Fig 4).

One of the respondents explained the situation, saying,

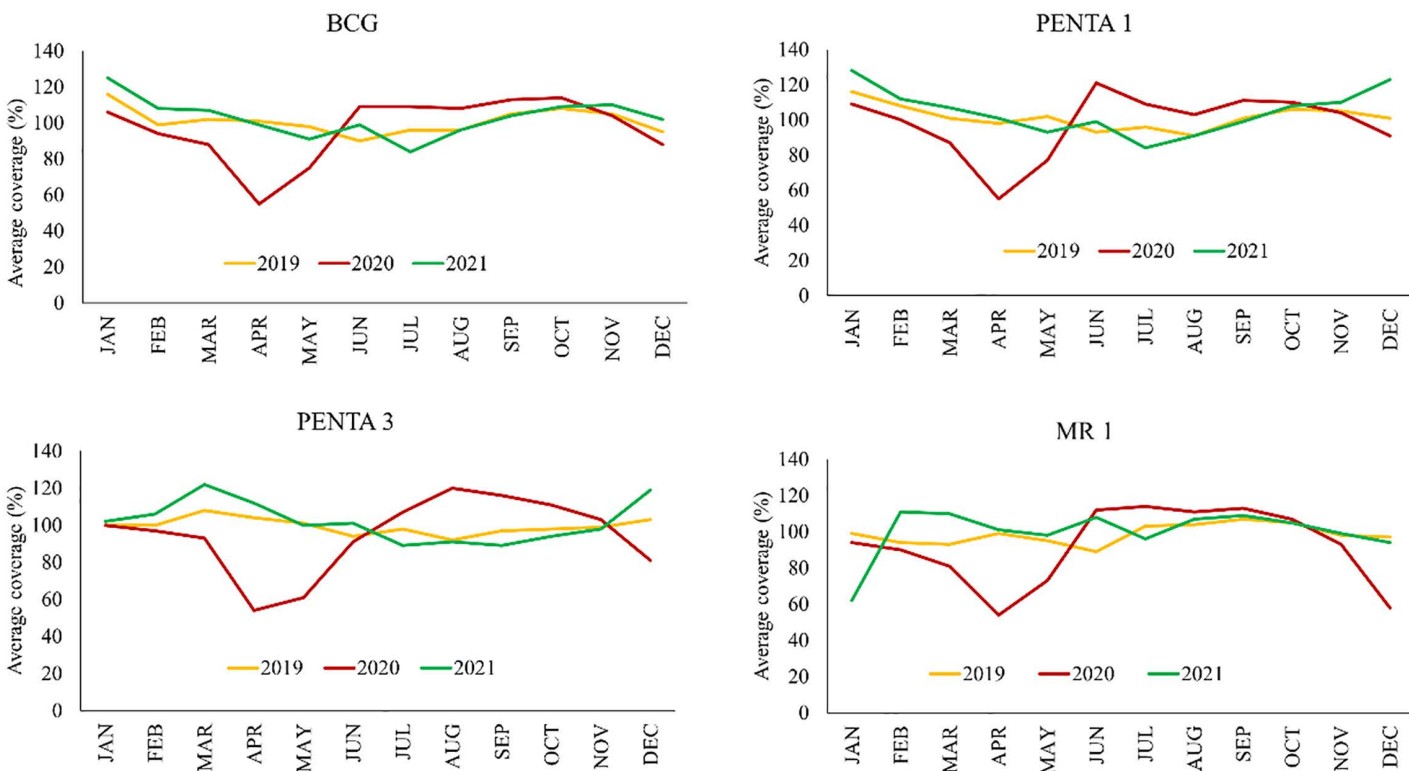

**Fig 2. Trend of BCG, PENTA 1, PENTA 3 and MR1 coverage, 2019-2021.**

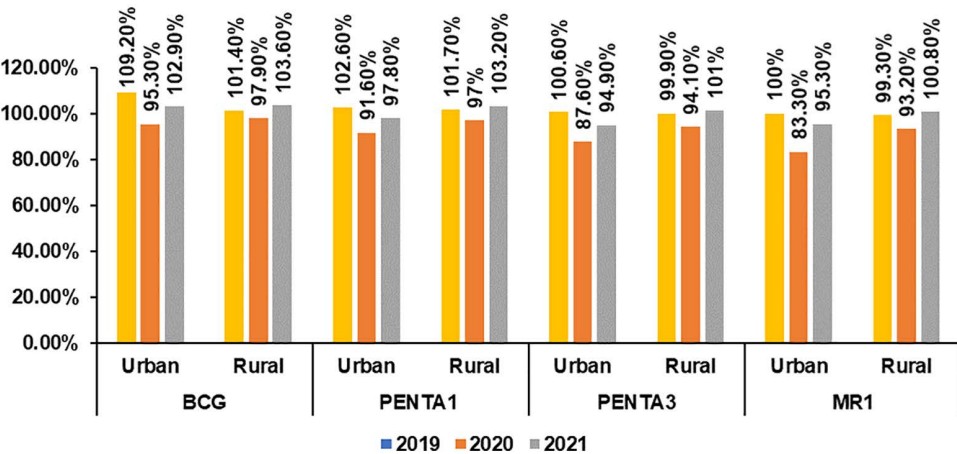

**Fig 3. Urban vs Rural BCG, PENTA 1, PENTA 3 & MR1 coverage in 2019-2021.**

*"The fact is that [VPD] surveillance was definitely lacking due to covid. It seemed that the field movement restriction and our coverage were a little down for VPD surveillance. Maybe we can catch up again after three months when the restrictions are a bit loose. Surveillance began in full swing, but still, our reporting rate of VPD surveillance was somewhat hampered."*

Quotation 5 [KII7, UH&FPO]

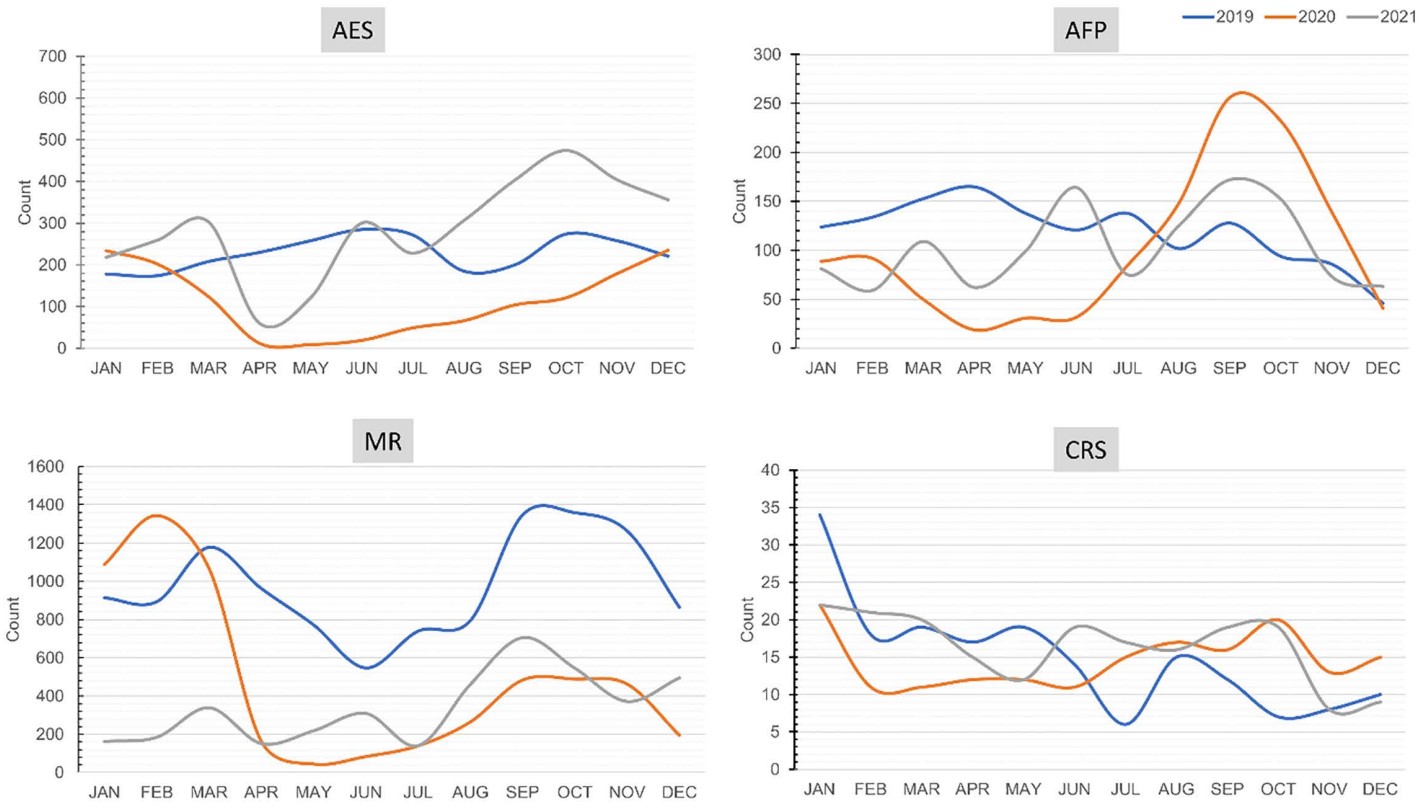

**Fig 4. VPD Surveillance in Bangladesh during 2019-2021.**

As with routine immunization, VPD surveillance was also affected during the COVID-19 pandemic, and it caught up after the initial phase of drop during the early days. According to the UH&FPO (Quote 4), the main reason for this drop in surveillance is credited to the COVID-19 movement restrictions that were strictly maintained during the early days of the pandemic in Bangladesh. Due to these restrictions, the collection and transportation of samples for testing became significantly difficult. One of the field-level personnel said,

*"[I]n the community level, it was difficult to collect samples due to COVID restrictions. If anyone was detected during that time, we treated him with local arrangements and cured him. It was tough to collect and transport the samples to Dhaka as there was almost no transportation available. Transport was completely shut down then with a government mandate."*

Quotation 6 [KII9, Medical Technologist]

In addition to the COVID-19 movement restriction, the study also found the issue of a lack of workforce at the field level, which significantly affected their capacity to maintain VPD surveillance during the pandemic. One of the health managers argued,

*"If I say in general about VPD, we still lack much staff in the field; due to this shortage, some of our operations are always interrupted… It appears that one person has to do the work of another, which hinders the total operation, and the monitoring cannot be maintained exactly the way it…if we do not have the staff, who are our main resource, then we cannot do the vaccination, surveillance, or whatever else we are [ought to be] doing in the field."*

Quotation 7 [KII8, UH&FPO]

In addition to the shortage of workforce at the field level, the study found that there is also a shortage of surveillance-related workforce in city corporations and at the national level. According to a respondent who is a high government official working in the health sector:

> "[W]e have a surveillance officer from WHO [now], but earlier we had two officers. Moreover, yesterday, I got to know that he [the current WHO surveillance officer] is also leaving us. So, I think the workforce should be strengthened. Moreover, here in the City Corporation, doctors or specialists [for surveillance] should be posted as employees of the city corporation. They should be given responsibility for the surveillance system, and it should be better. The WHO and UNICEF will also augment, but we should have our HR for surveillance."

Quotation 8 [KII1, Health Manager, Dhaka]

On top of the shortage of workforce in national, city corporation, and filed level, the existing VPD surveillance structure is dependent on WHO's Surveillance & immunization medical officer (SIMO) network.

However, VPD surveillance began to improve after the initial three to four months of the COVID-19 pandemic in Bangladesh (Quotation 5). The reason behind such a quick rebound is credited to utilizing online platforms during the COVID-19 period for monitoring and introducing refresher training to the field-level staff on VPD surveillance to improve their capacity.

## Discussion

Routine immunization and VPD surveillance systems are adopted to ensure the health and well-being of people in general and young people in particular [11,24]. With the emergence of the COVID-19 pandemic, these programs have faced an unprecedented challenge all over the world as they have suffered complete or partial suspension in the majority of countries all over the world [24]. The situation is even worse in the low-and-middle-income countries (LMICs), as most of the countries faced serious setbacks in their immunization activities during the early days of the COVID-19 pandemic [25]. Studies found that the LMICs have different challenges than the HICs in restoring their vaccination programs [26]. To explore the effects of COVID-19 on routine vaccination and VPD surveillance programs, this study explored the phenomenon in the context of Bangladesh, which belongs to the LMIC category.

Findings show that routine immunization and VPD surveillance coverage dropped significantly during March-April of 2020; similar drops were identified all over the world, including Europe, the USA, Latin America, and Asia [11]. In Pakistan, the vaccination coverage dropped 36%, and in Italy, it dropped to 33%, which is similar to Bangladesh, as the vaccination coverage dropped almost 40% at the same time [11,27]. However, Bangladesh was able to rebound and restore its coverage within June 2020 [28]. The inability to restore their vaccination coverage has led to multiple outbreaks, e.g., a polio outbreak in Niger, Chad, Ethiopia, Ghana, Pakistan and Afghanistan, all of which are LMICs [23]. Therefore, it is imperative to discuss the specific reasons why the vaccination and VPD surveillance coverage dropped and then rebounded in Bangladesh.

Drawing from Latour's Actor-Network theory, the routine immunization and VPD surveillance program in Bangladesh can be thought of as a network that is made of human and non-human actors, which includes but are not limited to the healthcare providers, government, non-government organizations, supra-government organizations, vaccines, pathogens, media, data, etc., and any disruption in the interaction between any actors will disrupt the whole network of vaccination and surveillance. In this network, every actor is important and dependent on another actor [29].

Borrowing from Loon, the coronavirus can be thought of as a vector, one of the three forces he mentioned to constitute the epidemic space, as it is an external force that created the new but temporary situation [30]. The epidemic space created by the coronavirus can be called the COVID-19 pandemic, in which the vaccination and VPD surveillance in Bangladesh have to function. However, in this epidemic space, the movements of people and goods are restricted, people

are scared and concerned about getting infected, vaccination campaigns are halted or cancelled, vaccinators are in fear of getting infected, and they are yet to have the instructions and training to carry out vaccination and surveillance in the epidemic space. Epidemic spaces with similar characteristics were created all over the world, as studies identified that routine vaccination was disrupted during COVID-19 pandemic due to (i) limits on movement and social isolation, (ii) fear and concerns of people for COVID-19 infection isolation, (iii) shifted priorities of medical professionals in favour of COVID-19 pandemic, (iv) logistical issues of delivering vaccines, (v) re-deployment and shortage of workforce, (vi) lack of personal protective equipment [7,9,26], (vii) cancelled routine programs in clinics and other places, (viii) misinformation and lack of knowledge regarding COVID-19 and immunization [7,9,24,26,31–33]

Findings suggest that Bangladesh had better vaccination coverage in the rural areas than in the urban areas, as its rural health structure is better than its urban structure. However, other LMICs, such as Pakistan, had the opposite picture, as the vaccination coverage in rural areas was less than in urban areas [34].

The study identified that there were three drop-rebound sessions in the EPI coverage and VPD surveillance. The first drop, i.e., March-April 2020, was the most serious in the coverage and surveillance as fear, uncertainty, and movement restrictions seriously limited these activities. This period represented the nadir of global vaccination coverage, with the lowest number of vaccine doses administered observed in April 2020, when 33% fewer DTP3 doses were administered globally [7]. However, the quantitative data shows another drop at the end of 2020, but the qualitative data argues that this drop is not an actual drop; instead, it was caused by the inconsistency of the DHIS2 data. During that time (December 2020-January 2021), there was a nationwide campaign on MR1 vaccination, which was scheduled in March 2020 but was delayed due to the COVID-19 pandemic [3]. In this campaign, a large number of children received MR1 vaccination, but this vaccination data was stored separately from the national data [35]. This separate approach to data management was shown to result in a drop in national data. The third drop in coverage and surveillance in the middle of 2021 was caused by the emergence of the delta variant [36] and the beginning of COVID-19 vaccination in the country [37].

The vaccination scenario of Bangladesh was also different from other LMICs in terms of maintenance of the immunization supply chain and logistics. Findings suggest that Bangladesh was able to maintain a robust supply chain throughout the pandemic by proper and timely mobilization of various local, national, and supra-national actors. However, studies found that maintaining the logistics and supply of vaccines was one of the significant challenges for most of the LMICs [25]. However, according to the findings, Bangladesh had suffered from a longstanding shortage of workforce during the COVID-19 pandemic, which is similar to other LMICs [25].

Another important factor in the drop in vaccination rates during the early days of the COVID-19 pandemic is self-preservation. From the findings, it is evident that people from both the demand side (the people who will get the vaccines) and the supply side (the people who will provide the vaccines) were in much fear and tension at the beginning of the pandemic. They perceived COVID-19 as a greater threat than that of the VPD and adopted various measures and strategies, such as staying inside homes and avoiding mass gatherings. Others lead to nonparticipation in routine immunization Parents and caregivers were reluctant to bring children to healthcare facilities due to concerns about exposure to the virus. This fear-driven behaviour was compounded by widespread movement restrictions and lockdown measures implemented by governments worldwide [38]. This phenomenon aligns with evidence from South Africa, where caregivers described caregiving decisions as "active decisions" to avoid non-essential services during the pandemic [39].Similarly, a study from Saudi Arabia reported that 35% of caregivers delayed vaccinations out of fear of infection [40].Together, these findings suggest that fear and self-preservation instinctively limited both demand for and provision of immunization services early in the pandemic, contributing to notable declines in vaccination coverage. Understanding their fear and concerns, the government and other relevant organizations have taken many initiatives, such as developing and distributing guidelines to maintain routine immunization during the pandemic situation, SBCC material, IPC training, and awareness campaigns. All these initiatives influenced peoples' self-preservation measures and strategies, as their risk perception toward COVID-19 and VPD shifted in favour of VPD.

## Study limitations

This study has several limitations. First, reliance on aggregated DHIS2 and EPI data may be affected by reporting errors, delays, or denominator inaccuracies, particularly in urban areas, occasionally yielding coverage estimates over 100%. Second, the qualitative component included only ten key informant interviews, which may not capture all perspectives across regions and administrative levels. Third, data quality disruptions during mass campaigns (e.g., MR vaccination) complicated comparisons with routine coverage. Finally, as the study focused on Bangladesh, findings may not be fully generalizable to other low- and middle-income countries. Despite these limitations, triangulating quantitative and qualitative data enhances the validity of our conclusions and offers insights into the resilience and vulnerabilities of the national immunization system.

## Conclusions

The COVID-19 pandemic substantially disrupted routine immunization and vaccine-preventable disease surveillance in Bangladesh, with three phases of decline and recovery between 2020 and 2021. Although coverage initially fell, most vaccines rebounded to pre-pandemic levels by late 2021, with rural areas recovering faster than urban settings due to stronger public infrastructure and community engagement. Surveillance of AFP, AES, and measles-rubella also declined early but improved with remote monitoring and infection-prevention training, while CRS surveillance remained stable. Persistent urban challenges, including high mobility, NGO dependence, and service mismatches, underscore the need for stronger infrastructure, adequate staffing, and flexible, technology-enabled surveillance to build resilience against future public health emergencies.

## Acknowledgments

We express our sincere gratitude to Md. Tahmid Hasan for his extensive contributions, particularly in supporting the qualitative component of this study. We are also thankful to Farha Nusrat Zahan and Md. Sahidul Islam for their essential roles in data analysis. Special thanks to Sudipta and Tamzid for their valuable input in shaping the discussion section. We gratefully acknowledge the support of the faculty and staff of the Institute of Global Health, University of Siena, Italy, for their academic guidance. We also extend our heartfelt appreciation to all Key Informant Interview (KII) participants for generously sharing their time and insights, which enriched the findings of this study.

## Author contributions

**Conceptualization:** Md. Tanvir Hossen, Md Sayik Bin Alam, Emanuele Montomoli, Md Foyjul Islam.

**Data curation:** Md. Tanvir Hossen.

**Formal analysis:** Md. Tanvir Hossen, Md Foyjul Islam.

**Investigation:** Md. Tanvir Hossen, Md Sayik Bin Alam.

**Methodology:** Md. Tanvir Hossen, Md Foyjul Islam.

**Project administration:** Md. Tanvir Hossen.

**Resources:** Md. Tanvir Hossen.

**Software:** Md. Tanvir Hossen.

**Validation:** Md. Tanvir Hossen.

**Visualization:** Md. Tanvir Hossen.

**Writing – original draft:** Md. Tanvir Hossen, Md Sayik Bin Alam, Md. Sahidul Islam, Emanuele Montomoli, Ralf Clemens, Sue Ann Costa Clemens, Mohammad Delwer Hossain Hawlader, Md Foyjul Islam.

**Writing – review & editing:** Md. Tanvir Hossen, Md Sayik Bin Alam, Md. Sahidul Islam, Emanuele Montomoli, Ralf Clemens, Sue Ann Costa Clemens, Mohammad Delwer Hossain Hawlader, Md Foyjul Islam.

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
