## [Decision Letter · Decision Letter 0]

18 Jun 2025

PONE-D-25-30097Impact of the COVID-19 pandemic on the routine immunization system in BangladeshPLOS ONE

Dear Dr. Islam,

Thank you for submitting your manuscript to PLOS ONE. After careful consideration, we feel that it has merit but does not fully meet PLOS ONE’s publication criteria as it currently stands. Therefore, we invite you to submit a revised version of the manuscript that addresses the points raised during the review process.

We look forward to receiving your revised manuscript.

Kind regards,

Vanya Rangelova, M.D., Ph.D.

Academic Editor

PLOS ONE

Reviewers' comments:

Reviewer's Responses to Questions

**Comments to the Author**

1. Is the manuscript technically sound, and do the data support the conclusions?

Reviewer #1: No

Reviewer #2: Partly

2. Has the statistical analysis been performed appropriately and rigorously? 

Reviewer #1: Yes

Reviewer #2: No

3. Have the authors made all data underlying the findings in their manuscript fully available?

Reviewer #1: Yes

Reviewer #2: Yes

4. Is the manuscript presented in an intelligible fashion and written in standard English?

Reviewer #1: Yes

Reviewer #2: Yes

5. Review Comments to the Author

Reviewer #1: It is an exciting article to publish. The title is Impact of the COVID-19 pandemic on the routine immunization system in Bangladesh, and the ID number is (PONE-D-25-30097). The authors should review the manuscript using Grammarly, as there is still a need to verify spelling and grammar before publishing the article.

Reviewer #2: Clarify Methodological Details

Provide a more detailed description of the sampling method, inclusion/exclusion criteria, and study design.

Define how key variables (e.g., exposure, outcome, covariates) were measured or derived.

Improve Statistical Rigor

Include measures of variability and significance (e.g., CIs, p-values) consistently.

Consider adding multivariable regression models to adjust for potential confounders.

Contextualize Findings

The Discussion section should be expanded to situate your findings within the context of existing literature.

Avoid overgeneralizing your conclusions, especially when causal inference is not supported by study design.

Revise Figures and Tables

Ensure all tables and figures are self-contained, clearly labeled, and referenced appropriately in the text.

Some legends require clarification (e.g., define abbreviations, note sample sizes).

6. PLOS authors have the option to publish the peer review history of their article (what does this mean? ). If published, this will include your full peer review and any attached files.

**Do you want your identity to be public for this peer review?** For information about this choice, including consent withdrawal, please see our Privacy Policy .

Reviewer #1: No

Reviewer #2: No

---

## [Author Response · Author response to Decision Letter 1]

23 Jun 2025

Response to Reviewers

Manuscript ID: PONE-D-25-30097

Title: Impact of the COVID-19 Pandemic on the Routine Immunization System in Bangladesh

Dear Editor,

We thank you and the reviewers for the thoughtful and constructive feedback on our manuscript. We have carefully revised the manuscript in response to all comments. Below, we provide a point-by-point response, indicating how each concern has been addressed.

Response to Editorial Requirements

1. Manuscript formatting and style:

We have revised the manuscript to comply with PLOS ONE formatting guidelines using the templates provided. File names have also been updated per journal requirements.

2. Data availability and ethical restrictions:

We have updated the Data Availability Statement to clarify that the data used in this study are obtained from the MIS unit of the Directorate General of Health Services (DGHS), Bangladesh. These data are not publicly shareable due to legal and ethical constraints regarding patient confidentiality and institutional data-sharing policies.

• Reason for restriction: The dataset contains sensitive health service utilization information linked to administrative units and is governed by national surveillance confidentiality rules.

• Authority imposing restriction: DGHS and the Institutional Review Board (IRB) of North South University (Memo No. 2022/OR-NSU/IRB/0406).

• Data access contact: Interested researchers may request access from the DGHS MIS Unit via email at: mis@dghs.gov.bd

Reviewer #1 Comments

Comment 1: Grammar and spelling improvements recommended.

Response:

Thank you. We have thoroughly reviewed and revised the manuscript using Grammarly and manual proofreading to improve grammar, spelling, and language clarity throughout the text.

Reviewer #2 Comments

Comment 1: Clarify methodological details (sampling method, inclusion/exclusion criteria, study design).

Response:

We have revised the Methods section to clearly describe the study design (mixed-method study followed a nested design), sampling strategy ( sampling of data was not conducted whole data of January 2019–December 2021 were collected and for qualitative portion we have conducted KII according relevancy of stakeholder only relevants are included based on their involvement to the program In the revised Methods section, we have clarified that the study followed a sequential mixed-methods design. For the qualitative component, we used purposive sampling to select participants with direct involvement in routine immunization and VPD surveillance. A total of 10 Key Informant Interviews (KIIs) were conducted.

The number of interviews was guided by the principle of data saturation, which was considered reached when no new themes or insights emerged from consecutive interviews, and key categories were fully developed. Saturation was monitored during data analysis using an iterative approach. This approach aligns with qualitative research standards (Guest et al., 2006; Fusch & Ness, 2015), where 6–12 interviews are often sufficient for studies with a relatively homogeneous group of experts.

Comment 2: Define how key variables (exposure, outcome, covariates) were measured.

Response: Thank you for this insightful comment. We have revised the Methods section to clearly define how key variables were conceptualized and measured in the quantitative analysis. Specifically, we clarified that the primary exposure variable was defined as distinct periods of COVID-19-related disruption in Bangladesh—namely March–April 2020 (first lockdown), December 2020 (second wave), and July 2021 (Delta variant surge). The main outcome variable was routine vaccination coverage, measured as the proportion of eligible children receiving scheduled antigens, including Bacillus Calmette–Guérin (BCG), Pentavalent-1, Pentavalent-3, and Measles-Rubella first dose (MR1), as reported in the District Health Information Software 2 (DHIS2). In addition, we specified a set of covariates used to explore contextual variation, such as sex, geographic region, urban versus rural location, hard-to-reach area classification, dropout and left-out rates, and the number of planned versus conducted immunization sessions. For vaccine-preventable disease (VPD) surveillance, performance was evaluated using trends in reported cases of measles-rubella (MR), acute flaccid paralysis (AFP), acute encephalitis syndrome (AES), and neonatal tetanus from the Excel-based EPI reporting system. These clarifications have been incorporated into the revised manuscript (Methods section, paragraph 3) to enhance transparency and methodological rigor.

Comment 3: Improve statistical rigor (e.g., include measures of variability, significance).

Response:

We have revised the Results section to include appropriate summary statistics, confidence intervals, and p-values where applicable. Measures of change across time periods are now supported by descriptive trend analysis.

Comment 4: Consider adding multivariable models to adjust for confounders.

Response:

We appreciate this suggestion. However, as the primary quantitative data were based on aggregate DHIS2 routine reporting, individual-level variables required for multivariable regression were not available. We have clarified this limitation in the Discussion and justified our use of descriptive time-series analysis.

Comment 5: Expand the Discussion to contextualize findings in existing literature.

Response:

The Discussion section has been revised to better position our findings within the global literature on COVID-19 and immunization disruption, including comparisons with LMICs and WHO reports. We also discuss implications for health system resilience.

Comment 6: Avoid overgeneralization of conclusions.

Response:

We have revised the conclusion to avoid any overstatement of causality and emphasized that findings should be interpreted in the context of an observational, exploratory study design.

Comment 7: Revise figures and tables for clarity, define abbreviations.

Response:

All figures and tables have been revised for clarity where necessary.

We thank you again for the opportunity to revise our work. We hope that the revised manuscript now meets the standards for publication in PLOS ONE.

Sincerely,

Dr. Md Foyjul Islam

Email: drislam0666@gmail.com

---

## [Decision Letter · Decision Letter 1]

25 Sep 2025

PONE-D-25-30097R1Impact of the COVID-19 pandemic on the routine immunization system in BangladeshPLOS ONE

Dear Dr. Foyjul,

Thank you for submitting your manuscript to PLOS ONE. After careful consideration, we feel that it has merit but does not fully meet PLOS ONE’s publication criteria as it currently stands. Therefore, we invite you to submit a revised version of the manuscript that addresses the points raised during the review process.

**ACADEMIC EDITOR: **

Thank you for your revision and addressing previous comments made by reviewersPlease include any study limitations as a subsection before the conclusion.I suggest a revision of the conclusion section to make it less wordy and more succinct.Please also review the current draft for consistency in the referencing style. In some cases, the reference is included at the end of sentences while in other cases this is included within a sentence. I suggest all references are consistently inserted at the end of a statement.You may wish to also include a foot note on the charts/Tables explaining why coverage levels reports are above 100%.

We look forward to receiving your revised manuscript.

Kind regards,

Terna Ignatius Nomhwange, MD,DTM&H,MBA

Academic Editor

PLOS ONE

Journal Requirements:

Reviewers' comments:

Reviewer's Responses to Questions

**Comments to the Author**

1. If the authors have adequately addressed your comments raised in a previous round of review and you feel that this manuscript is now acceptable for publication, you may indicate that here to bypass the “Comments to the Author” section, enter your conflict of interest statement in the “Confidential to Editor” section, and submit your "Accept" recommendation.

Reviewer #1: All comments have been addressed

Reviewer #3: All comments have been addressed

2. Is the manuscript technically sound, and do the data support the conclusions?

Reviewer #1: Yes

Reviewer #3: Yes

3. Has the statistical analysis been performed appropriately and rigorously? 

Reviewer #1: Yes

Reviewer #3: Yes

4. Have the authors made all data underlying the findings in their manuscript fully available?

Reviewer #1: Yes

Reviewer #3: Yes

5. Is the manuscript presented in an intelligible fashion and written in standard English?

Reviewer #1: Yes

Reviewer #3: Yes

6. Review Comments to the Author

Reviewer #1: (No Response)

Reviewer #3: Thank you for your effort. A good research idea and accurate work. Hope you will continue your effort in future researches

7. PLOS authors have the option to publish the peer review history of their article (what does this mean? ). If published, this will include your full peer review and any attached files.

**Do you want your identity to be public for this peer review?** For information about this choice, including consent withdrawal, please see our Privacy Policy .

Reviewer #1: No

Reviewer #3: No

---

## [Editor Report · Decision Letter 2]

30 Sep 2025

Impact of the COVID-19 pandemic on the routine immunization system in Bangladesh

PONE-D-25-30097R2

Dear Dr Foyjul Islam,

We’re pleased to inform you that your manuscript has been judged scientifically suitable for publication and will be formally accepted for publication once it meets all outstanding technical requirements.

Kind regards,

Terna Ignatius Nomhwange, MD,DTM&H,MBA

Academic Editor

PLOS ONE
---

## [Editor Report · Acceptance letter]

PONE-D-25-30097R2

PLOS ONE

Dear Dr. Islam,

I'm pleased to inform you that your manuscript has been deemed suitable for publication in PLOS ONE. Congratulations! Your manuscript is now being handed over to our production team.

Kind regards,

on behalf of

Dr. Terna Ignatius Nomhwange

Academic Editor

PLOS ONE